# NOT IN SYNC: UNVEILING TEMPORAL BIAS IN AUDIO CHAT MODELS

## ABSTRACT

Large Audio Language Models (LALMs) are increasingly applied to audio understanding and multimodal reasoning, yet their ability to locate when events occur remains underexplored. We present the first systematic study of temporal bias in LALMs, revealing a key limitation in their timestamp prediction. For example, when asked "At which second does the lecturer introduce the key formula?", models often predict timestamps that are consistently earlier or later than the ground truth. Through controlled experiments on timestamped datasets, we find that temporal bias (i) is prevalent across datasets and models, (ii) increases with audio length—even accumulating to tens of seconds in extended recordings, and (iii) varies across event types and positions. We quantify this effect with the Temporal Bias Index (TBI), measuring systematic misalignment in predicted event timings, and complement it with a visualization framework. Our findings highlight a fundamental limitation in current LALMs and call for the development of temporally robust architectures.

## 1 INTRODUCTION

Large Audio-Language Models (LALMs) have recently shown remarkable progress in understanding, reasoning, and generating language conditioned on audio signals. Beyond semantic comprehension, however, real-world applications often demand temporal awareness: users do not merely ask **what** is said, but also **when** it occurs. For instance, in lecture indexing, users may ask: *"At which second does the lecturer introduce the key formula?"* Similarly, in political debates or musical performances, knowing the temporal location of an event is as crucial as recognizing its content. Despite the importance of temporal grounding, the ability of LALMs to perceive event timing remains underexplored.

We identify a fundamental issue in this dimension, which we term **temporal bias**. This refers to the systematic tendency of LALMs to misplace audio events along the time axis. Figure 1 provides a stark illustration of this phenomenon through a detailed case study. The figure demonstrates how a state-of-the-art LALM can successfully summarize the key events in an audio clip—a testament to its semantic understanding. This semantic competence is further corroborated by its internal attention mechanism, which correctly focuses on the relevant audio segments (right panel). However, the same model exhibits a severe temporal bias, systematically misreporting the timestamps for these very events by several seconds. This reveals a critical dissociation between the model's robust semantic comprehension and its fragile temporal localization abilities, undermining its reliability in precision-critical tasks.

To investigate this phenomenon, we design a series of controlled experiments. The setup manipulates three factors: (1) *sequence length*, where identical events are embedded into audios of varying total durations; (2) *event duration*, where both short and extended occurrences are compared; and (3) *event position*, where events appear at the beginning, middle, or end of the audio. By comparing model predictions with ground-truth timestamps derived from a strong ASR baseline, we quantify error through mean absolute deviation and further propose a **Temporal Bias Index (TBI)** to capture directional tendencies.

Our findings reveal a non-trivial pattern. LALMs do not simply anticipate events earlier than their ground truth, as might be intuitively assumed. Instead, the bias fluctuates across contexts: shorter audios often induce overestimation of event timing, longer audios amplify underestimation, and

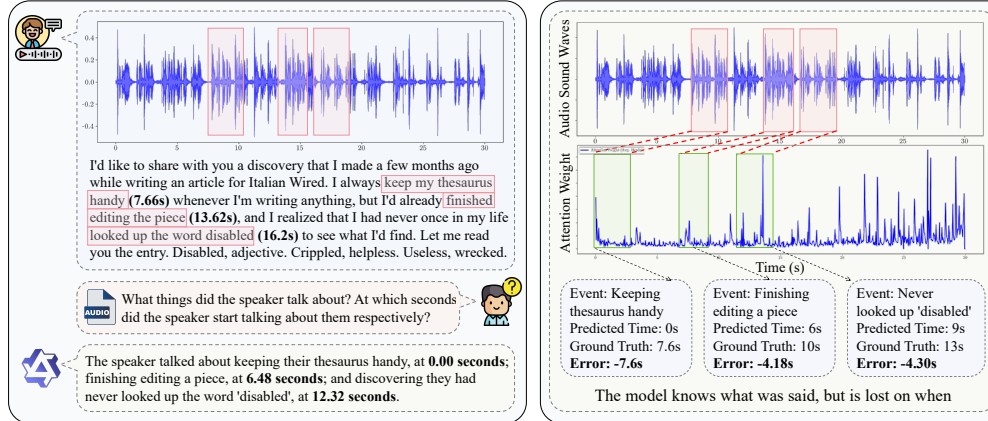

Figure 1: **Semantic Success, Temporal Failure.** A case study of an LALM tasked with summarizing and timestamping events. **(Left)** The model produces a semantically correct summary with highly inaccurate timings. **(Right)** Internal attention correctly focuses on the relevant audio segments, yet large temporal errors persist, highlighting the paradox: the model knows *what* was said, but is lost on *when*.

positional effects follow a U-shaped curve, with boundary events more error-prone than central ones. These results suggest that temporal misalignment is systematic yet heterogeneous, reflecting deeper architectural limitations in how LALMs encode temporal structure.

This paper presents the first systematic study dedicated to characterizing and quantifying temporal bias in LALMs. Our contributions are as follows:

- Through extensive experiments, we uncover that LALMs exhibit systematic but non-uniform biases: they neither universally anticipate nor universally delay events, but shift timings depending on contextual factors.
- We design a systematic evaluation framework across sequence length, event duration, and event position, and quantify the bias with a novel **Temporal Bias Index (TBI)**.
- We provide interpretability analyses that connect these biases to attention and positional encoding compression, offering insights for future temporally robust model design.

## 2 TEMPORAL BIAS PHENOMENON

Large audio language models are expected to not only understand what is happening in an audio stream but also to identify when events occur. However, our initial experiments reveal a systematic temporal bias: when asked to report the timing of events, the models struggle with temporal precision, often producing timestamps that systematically precede the ground truth. For example, when a user listens to a classroom recording and asks the model when the teacher introduces a formula, the answer tends to appear seconds earlier than the annotated moment. Similar patterns are observed in conversational audio, music, and environmental sounds. This systematic bias motivates a deeper investigation into how such models perceive time.

### 2.1 QUANTIFYING TEMPORAL MISALIGNMENT

To systematically measure the degree of temporal misalignment, we introduce metrics that capture both the direction and magnitude of the error. We formalize the task of temporal reasoning in audio language models as follows. Given an audio segment $x$ and a target event $e$, the model is prompted to output the predicted onset time $\hat{t}_e$.

**Temporal Bias Index** We define the Temporal Bias Index (TBI) as the average signed error between predicted and ground truth event onsets:

Table 1: Initial evidence of temporal bias across tasks. Negative TBI indicates early predictions. PretrainedSED serves as a supervised baseline. Values are reported as mean ± std in seconds.

| Model | STARSS22 | | Tedlium3 | | Music | |
|---|---|---|---|---|---|---|
| | **TBI** | **MAE** | **TBI** | **MAE** | **TBI** | **MAE** |
| Voxtral-Mini-3B-2507 | $-5.3_{\pm0.48}$ | $6.1_{\pm0.52}$ | $-4.7_{\pm0.42}$ | $5.9_{\pm0.49}$ | $-6.1_{\pm0.55}$ | $6.8_{\pm0.60}$ |
| Qwen2-Audio-7B-Instruct | $-4.8_{\pm0.45}$ | $5.6_{\pm0.50}$ | $-5.2_{\pm0.49}$ | $6.3_{\pm0.55}$ | $-5.9_{\pm0.53}$ | $6.5_{\pm0.58}$ |
| Kimi-Audio-7B-Instruct | $-6.0_{\pm0.52}$ | $6.9_{\pm0.63}$ | $-5.4_{\pm0.50}$ | $6.4_{\pm0.57}$ | $-6.8_{\pm0.61}$ | $7.3_{\pm0.68}$ |
| Aero-1-Audio | $-5.7_{\pm0.51}$ | $6.5_{\pm0.59}$ | $-6.2_{\pm0.58}$ | $7.1_{\pm0.66}$ | $-6.5_{\pm0.60}$ | $7.0_{\pm0.64}$ |
| PretrainedSED (baseline) | $-0.7_{\pm0.12}$ | $1.4_{\pm0.21}$ | $-0.5_{\pm0.09}$ | $1.2_{\pm0.18}$ | $-0.9_{\pm0.15}$ | $1.6_{\pm0.25}$ |

$$\text{TBI} = \frac{1}{N} \sum_{i=1}^{N} \left( \hat{t}_i - t_i \right),$$

where $N$ is the number of valid predictions, $t_i$ is the ground truth onset time, and $\hat{t}_i$ the model prediction. A negative TBI indicates systematic early bias (the model anticipates events). A positive TBI indicates systematic late bias (the model lags behind).

**Mean Absolute Error(MAE)**    To quantify the overall deviation regardless of direction, we also report the MAE on the start timestamps:

$$\text{MAE}_{\text{start}} = \frac{1}{N} \sum_{i=1}^{N} |\text{pred\_start}_i - \text{gt\_start}_i|,$$

where $N$ is the total number of evaluated events. Lower values indicate more precise temporal predictions.

## 2.2 Empirical Evidence of a Systematic Bias

We first examine whether temporal bias is consistently present across diverse conditions. Table 1 summarizes results on multiple tasks: acoustic events from STARSS22 (Politis et al., 2022b), conversational speech from Tedlium3, and music clips with annotated onsets. Across all settings, LALMs reveals a consistent bias pattern, highlighting a previously underexplored temporal sensitivity in audio language models.

In contrast, a supervised sound event detection model (PretrainedSED) maintains near-zero bias, demonstrating that such systematic temporal bias is unique to LALMs rather than an artifact of the datasets.

These findings establish temporal bias as a pervasive phenomenon in LALMs. While conventional event detection models are not entirely immune to temporal errors, this bias is significantly more pronounced and systematic in LALMs. This motivates a deeper exploration into how such bias evolves with input length, event characteristics, and other factors, which we examine in Section **??**.

## 3 Systematic Analysis of Temporal Bias

Having established in the previous section that temporal bias is a pervasive and systematic weakness in LALMs, we now shift our focus from detection to analysis. The initial findings motivate critical questions: How does the scale of this bias change as the context grows? Are certain types of events harder to locate in time than others? Does the model's performance vary depending on when an event occurs?

To answer these questions, this section presents a series of controlled experiments designed to isolate and examine the key factors influencing temporal bias. We will systematically analyze the impact of audio length, event duration, and event position on the temporal reasoning capabilities of LALMs.

## 3.1 EXPERIMENTAL SETUP

**Datasets and Experimental Design**   We conduct our experiments on the STARSS22 dataset, an authoritative benchmark containing a variety of real-world acoustic scenes with detailed event annotations. To systematically dissect temporal bias, our study is structured around three controlled variables. First, to analyze the impact of context size, we crop audio into segments of varying durations, ranging from 5s to 120s. For each segment, we randomly select up to four distinct event categories for querying to ensure consistent evaluation. Second, to understand how event characteristics affect localization, we create both short-duration ($< 3\,$s) and long-duration (7–10s) versions of events. Third, to probe for positional sensitivity, we divide each audio window into five uniform segments and place target events within them to measure performance as a function of temporal position. We also control for factors like background noise and event overlap to assess model robustness under realistic conditions.

**Models.**   Our evaluation features four state-of-the-art Large Audio Language Models (LALMs). These models are Voxtral-Mini-3B-2507 (Liu et al., 2025), a model with strong temporal understanding capabilities, Qwen2-Audio-7B-Instruct (Chu et al., 2024), an instruction-tuned multimodal model, as well as Kimi-Audio-7B-Instruct (KimiTeam et al., 2025) and Aero-1-Audio (Li et al., 2025). For comparison, we include PretrainedSED (Schmid et al., 2025), a Pretrained Sound Event Detection (SED) model, as a non-LALM baseline, which is trained with supervision to provide frame-level timestamps and allows for a direct assessment of the bias unique to LALMs.

**Implementation and Data Processing**   All models are evaluated in a zero-shot, unified audio question-answering (AQA) format, where the input consists of an audio clip and a textual query describing the event. We preprocess the STARSS22 dataset by merging adjacent frame-level annotations into contiguous events and segmenting recordings into fixed, non-overlapping windows. To isolate the task of temporal localization from event detection, models are provided with the oracle set of event classes present in each segment and are instructed to predict the earliest onset time for each. All experiments are conducted on GPUs with 40GB of memory, and prompts are standardized to minimize induced bias. Invalid outputs, such as empty or non-parsable answers, are excluded from our analysis.

## 3.2 RESULTS AND ANALYSIS

In this section, we present results from three controlled experiments: the effect of audio length, event duration, and event position on temporal reasoning performance.

### 3.2.1 EFFECT OF AUDIO LENGTH

Table 2 reports the Mean Absolute Error (MAE) across varying audio window lengths on the STARSS22 dataset (Sony and TAU subsets). To mitigate the impact of spurious predictions when the LALMs fail to correctly identify event classes, we conducted a prior experiment to first identify the event classes detected by the LALMs before predicting start times. Based on this, we divide predictions into three categories: *Correct* (event class identified with accurate timing), *Incorrect* (event class not detected), and *Overall* (all valid predictions regardless of event class identification).

Before analyzing LALMs in detail, we compare with a traditional audio event detection model PretrainedSED. Unlike LALMs, though even these models show degradation with increased context length, their temporal bias remains relatively stable across increasing context lengths, indicating that the pronounced length-dependent bias is unique to LALMs rather than an inherent property of audio modeling. Since PretrainedSED does not suffer from spurious predictions, it is evaluated using only the *Overall* category.

Several consistent patterns emerge. First, the magnitude of prediction bias grows dramatically with longer input windows, reaching more than 21 seconds at 120-second contexts. Second, the overall MAE escalates non-linearly with audio length, underscoring a systematic deterioration in temporal perception under extended contexts. Third, variance also increases with context size, implying that predictions not only become biased but also less stable and reliable.

Table 2: MAE across different audio window lengths on STARSS22 dataset. Values represent mean temporal bias in seconds with estimated standard deviation in parentheses.

| Model | Category | 5s | 30s | 60s | 90s | 120s |
|---|---|---|---|---|---|---|
| Voxtral-Mini-3B-2507 (LALM) | Correct | $0.85_{(\pm 0.05)}$ | $4.33_{(\pm 0.26)}$ | $7.17_{(\pm 1.85)}$ | $11.94_{(\pm 1.84)}$ | $22.62_{(\pm 1.92)}$ |
| | Incorrect | $0.91_{(\pm 0.00)}$ | $7.39_{(\pm 0.32)}$ | $15.73_{(\pm 1.75)}$ | $22.24_{(\pm 2.79)}$ | $29.66_{(\pm 3.35)}$ |
| | Overall | $0.90_{(\pm 0.00)}$ | $6.58_{(\pm 0.27)}$ | $12.69_{(\pm 1.80)}$ | $18.32_{(\pm 1.86)}$ | $23.45_{(\pm 2.67)}$ |
| Qwen2-Audio-7B-Instruct (LALM) | Correct | $0.54_{(\pm 0.10)}$ | $4.96_{(\pm 0.11)}$ | $11.09_{(\pm 0.57)}$ | $17.64_{(\pm 2.01)}$ | $24.04_{(\pm 3.41)}$ |
| | Incorrect | $1.58_{(\pm 0.00)}$ | $5.27_{(\pm 0.55)}$ | $13.43_{(\pm 0.65)}$ | $18.91_{(\pm 1.70)}$ | $45.44_{(\pm 2.05)}$ |
| | Overall | $1.50_{(\pm 0.03)}$ | $5.12_{(\pm 0.03)}$ | $12.23_{(\pm 2.02)}$ | $18.12_{(\pm 1.89)}$ | $30.80_{(\pm 2.42)}$ |
| Aero-1-Audio (LALM) | Correct | $2.18_{(\pm 0.04)}$ | $6.10_{(\pm 0.31)}$ | $11.23_{(\pm 0.90)}$ | $22.21_{(\pm 1.56)}$ | $25.05_{(\pm 1.77)}$ |
| | Incorrect | $2.62_{(\pm 0.16)}$ | $6.32_{(\pm 0.42)}$ | $15.12_{(\pm 1.16)}$ | $23.32_{(\pm 1.96)}$ | $28.96_{(\pm 2.33)}$ |
| | Overall | $2.51_{(\pm 0.06)}$ | $6.24_{(\pm 0.36)}$ | $13.61_{(\pm 1.04)}$ | $23.02_{(\pm 1.82)}$ | $27.08_{(\pm 2.03)}$ |
| Kimi-Audio-7B-Instruct (LALM) | Correct | $0.55_{(\pm 0.05)}$ | $2.10_{(\pm 0.22)}$ | $8.23_{(\pm 0.74)}$ | $18.34_{(\pm 1.65)}$ | $23.60_{(\pm 1.78)}$ |
| | Incorrect | $0.60_{(\pm 0.06)}$ | $3.30_{(\pm 0.33)}$ | $16.41_{(\pm 1.07)}$ | $34.75_{(\pm 2.11)}$ | $35.75_{(\pm 2.42)}$ |
| | Overall | $0.57_{(\pm 0.05)}$ | $2.67_{(\pm 0.28)}$ | $10.48_{(\pm 0.93)}$ | $21.90_{(\pm 1.96)}$ | $25.06_{(\pm 2.02)}$ |
| PretrainedSED (Non-LALM) | Overall | $0.20_{(\pm 0.00)}$ | $1.16_{(\pm 0.00)}$ | $1.72_{(\pm 0.02)}$ | $3.10_{(\pm 0.05)}$ | $2.87_{(\pm 0.02)}$ |

The severity of degradation differs across models. For Voxtral-Mini-3B-2507, the bias grows from 0.85 seconds at 5-second windows to 22.62 seconds at 120-second windows, which indicates a 26-fold increase that far exceeds linear scaling. Similarly, Qwen2-Audio shows bias escalation from 0.54 to 24.04 seconds, representing a 45-fold deterioration. Aero-1-Audio demonstrates the most severe temporal degradation, with bias reaching 25.05 seconds in the longest contexts. We can observe that longer contexts introduce compounding difficulties for precise temporal reasoning, potentially due to information dilution and memory span limitations. The phenomenon is universal, suggesting a fundamental architectural issue rather than a model-specific artifact.

The standard deviations increase proportionally with bias magnitude, indicating that temporal prediction becomes not only systematically biased but also increasingly variable and unreliable as audio length grows. This dual degradation—both systematic bias and increased variance—suggests fundamental limitations in how current audio language models process temporal information across extended contexts.

These findings establish temporal bias as a critical bottleneck for deploying audio language models in applications requiring precise event timing, such as lecture indexing, meeting analysis, or multimedia retrieval systems. The detailed breakdown of early versus late predictions and additional statistical analyses are provided in Appendix A (Table 5).

### 3.2.2 EFFECT OF EVENT DURATION

To investigate how temporal localization is influenced by the intrinsic properties of sound events, we designed a controlled experiment focusing on two key factors: event duration and event nature. We categorized events into two distinct types: **sustained events**, which have a continuous acoustic presence (e.g., 'Female Speech', 'Male Speech'), and **transient events**, which are brief and often percussive (e.g., 'Clap', 'Knock', 'Laughter').

To create a fair comparison, we prepared two duration-based versions for each event category. For sustained events, we used their naturally long recordings (7–10 s) and created short versions ($< 3$ s) by truncating the audio. Conversely, for transient events, we took their naturally short recordings ($< 3$ s) and generated long versions (7–10 s) by seamlessly looping the event audio. This methodology allows us to isolate the impact of duration while also comparing performance across fundamentally different acoustic patterns. The results are presented in Table 3.

Our analysis reveals two primary findings. First, a universal trend confirmed by the positive $\Delta$ values is that **longer event durations consistently degrade temporal localization performance for both models across all categories**. The mean absolute deviation increases significantly when models are tasked with localizing events within a longer time frame.

Table 3: Effect of event duration on mean absolute deviation (in seconds). For each model, the increase in deviation from short to long events is quantified in the $\Delta$ (Long-Short) column. A positive $\Delta$ indicates performance degradation on longer events.

| | Voxtral-Mini-3B | | | Qwen2-Audio-7B | | |
|---|---|---|---|---|---|---|
| Category | Short (<3s) | Long (7-10s) | $\Delta$ (L-S) | Short (<3s) | Long (7-10s) | $\Delta$ (L-S) |
| Female Speech | 3.253 | 5.088 | +1.835 | 2.981 | 4.887 | +1.906 |
| Male Speech | 3.237 | 4.342 | +1.105 | 3.012 | 4.115 | +1.103 |
| Laughter | 4.746 | 6.769 | +2.023 | 4.498 | 6.131 | +1.633 |
| Clap | 5.352 | 8.222 | +2.870 | 4.987 | 7.502 | +2.515 |
| Knock | 6.255 | 8.356 | +2.101 | 5.913 | 7.545 | +1.632 |
| Door Closing | 4.710 | 7.876 | +3.166 | 4.501 | 7.155 | +2.654 |
| Footsteps | 4.012 | 6.500 | +2.488 | 0.806 | 4.019 | +3.213 |
| Telephone | 4.727 | 6.920 | +2.193 | 1.127 | 3.720 | +2.593 |
| Bell | 6.200 | 8.956 | +2.756 | 5.875 | 7.910 | +2.035 |

Second, and more critically, **the nature of the event itself is a dominant factor in localization accuracy**. Even in the short-duration case, models perform markedly better on sustained speech events (deviations around 3.0–3.2 s) than on transient events. For instance, percussive sounds like 'Knock' and 'Bell' exhibit much higher initial deviations (5.9–6.2 s), suggesting that models are inherently less precise at pinpointing brief, sharp acoustic signals compared to continuous ones.

Furthermore, the performance degradation ($\Delta$) caused by increased duration is far more severe for transient events. While the error for 'Male Speech' increases by only +1.1 s, the deviation for 'Door Closing' and 'Clap' surges by as much as +3.166 s and +2.870 s for Voxtral, respectively. This suggests that extending transient events via repetition introduces significant temporal ambiguity, making it challenging for models to identify a single, representative timestamp. In comparing the two models, Qwen2 generally exhibits slightly lower deviations and is marginally more robust to the increase in duration, though it fundamentally shares the same limitations as Voxtral. These results underscore that both event duration and its intrinsic acoustic nature are critical variables affecting the temporal reasoning capabilities of LALMs.

### 3.2.3 Effect of Event Position in Audio

Building on our finding that event duration and nature impact temporal localization, we further probe whether the event's *position* within the audio clip introduces an additional layer of bias. An event occurring at the beginning of a long audio stream might be processed differently from one occurring near the end. To investigate this, we placed sound events at five equidistant relative positions within audio clips of varying lengths, from the start (Position 1) to the end (Position 5).

The results, presented in Table 4, reveal distinct positional biases for each model. Voxtral-Mini-3B consistently demonstrates a symmetric, U-shaped error pattern. For instance, in 60-second audio, the mean absolute deviation is highest at the start (23.94 s) and end (30.30 s), while dropping to its minimum in the central segment (14.70 s). This suggests that the model is least confident when localizing events near the temporal boundaries of the context it processes. In contrast, Qwen2-Audio-7B exhibits a more asymmetric and varied bias. In shorter clips (e.g., 10 s), the error progressively increases from the beginning to the end. In longer clips, while the pattern is less monotonic, a significant error accumulation is often observed in the final segment, indicating a strong sensitivity to events occurring late in the audio stream.

While Table 4 shows the absolute deviation, comparing these values across different audio lengths can be challenging, as longer clips naturally permit larger errors. To provide a fair, scale-invariant comparison, we also calculated the normalized mean absolute deviation by dividing the deviation by the total audio length. This metric, visualized in Figure 2, isolates the effect of relative position. The figure corroborates our findings, making Voxtral's U-shaped bias and Qwen2's end-of-clip sensitivity even more apparent. These findings conclusively establish that an event's position is a third critical factor, alongside its duration and nature, that significantly influences the temporal reasoning capabilities of LALMs.

Table 4: Effect of event position on temporal localization, showing the mean absolute deviation in seconds. The audio is divided into five equal segments (1=Start, 5=End). The primary value in each cell is the mean absolute deviation, with its standard deviation shown in subscript, e.g., $_{(\pm\sigma)}$.

| Model | Audio Length | Event Position within Audio (Start → End) | | | | |
|---|---|---|---|---|---|---|
| | | **1** | **2** | **3** | **4** | **5** |
| Voxtral-Mini-3B | 10s | $1.663_{(\pm 1.637)}$ | $0.811_{(\pm 0.819)}$ | $0.428_{(\pm 0.428)}$ | $0.612_{(\pm 0.612)}$ | $1.363_{(\pm 1.363)}$ |
| | 20s | $4.108_{(\pm 4.108)}$ | $2.696_{(\pm 2.696)}$ | $1.991_{(\pm 1.991)}$ | $2.978_{(\pm 2.978)}$ | $4.731_{(\pm 4.731)}$ |
| | 30s | $7.801_{(\pm 7.801)}$ | $4.954_{(\pm 4.954)}$ | $3.102_{(\pm 3.102)}$ | $5.922_{(\pm 5.922)}$ | $9.605_{(\pm 9.605)}$ |
| | 40s | $11.88_{(\pm 11.88)}$ | $8.318_{(\pm 8.318)}$ | $6.166_{(\pm 6.166)}$ | $9.205_{(\pm 9.205)}$ | $14.40_{(\pm 14.40)}$ |
| | 50s | $17.70_{(\pm 17.70)}$ | $13.20_{(\pm 13.20)}$ | $9.953_{(\pm 9.953)}$ | $13.95_{(\pm 13.95)}$ | $22.15_{(\pm 22.15)}$ |
| | 60s | $23.94_{(\pm 23.94)}$ | $18.96_{(\pm 18.96)}$ | $14.70_{(\pm 14.70)}$ | $19.80_{(\pm 19.80)}$ | $30.30_{(\pm 30.30)}$ |
| Qwen2-Audio-7B-Instruct | 10s | $1.240_{(\pm 1.240)}$ | $1.704_{(\pm 1.704)}$ | $2.951_{(\pm 2.951)}$ | $4.269_{(\pm 4.269)}$ | $5.404_{(\pm 5.404)}$ |
| | 20s | $8.149_{(\pm 8.149)}$ | $6.136_{(\pm 6.136)}$ | $4.283_{(\pm 4.283)}$ | $2.736_{(\pm 2.736)}$ | $3.320_{(\pm 3.320)}$ |
| | 30s | $8.384_{(\pm 8.384)}$ | $5.447_{(\pm 5.447)}$ | $3.887_{(\pm 3.887)}$ | $6.393_{(\pm 6.393)}$ | $10.35_{(\pm 10.35)}$ |
| | 40s | $13.74_{(\pm 13.74)}$ | $10.99_{(\pm 10.99)}$ | $6.232_{(\pm 6.232)}$ | $7.300_{(\pm 7.300)}$ | $6.770_{(\pm 6.770)}$ |
| | 50s | $16.87_{(\pm 16.87)}$ | $9.290_{(\pm 9.290)}$ | $7.614_{(\pm 7.614)}$ | $6.290_{(\pm 6.290)}$ | $8.490_{(\pm 8.490)}$ |
| | 60s | $9.105_{(\pm 9.105)}$ | $8.929_{(\pm 8.929)}$ | $7.795_{(\pm 7.795)}$ | $18.81_{(\pm 18.81)}$ | $22.86_{(\pm 22.86)}$ |

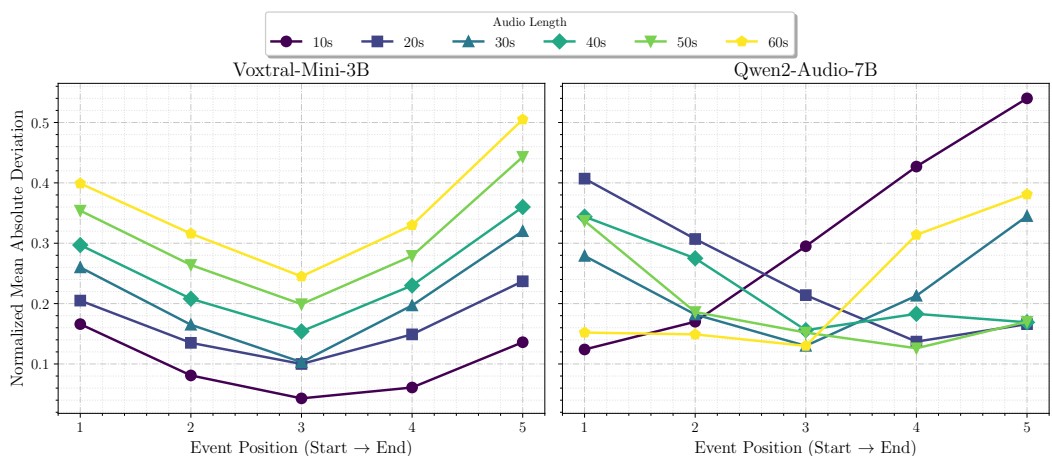

Figure 2: Normalized mean absolute deviation by event position.

## 4 INTERPRETABILITY ANALYSIS

The preceding sections have empirically established that Large Audio Language Models (LALMs) exhibit a pervasive and complex temporal bias. Having quantified *what* this bias is, we now turn to the question of *why* it occurs. What are the underlying mechanisms that lead to this phenomenon? Why do these models demonstrate a sophisticated semantic understanding, yet systematically fail at the seemingly simpler task of temporal localization? To answer these questions, this section presents an interpretability analysis designed to deconstruct the model's internal reasoning process. We conduct a case study on the Qwen2-Audio model, tracing the layer-wise evolution of its cross-modal attention while processing a 30-second audio clip from a TED talk.

Figure 3 provides a panoramic view of this internal process, visualizing the attention distribution across six representative decoder layers. The figure reveals a striking and systematic transformation in how the model attends to the audio over its depth. Our central hypothesis, supported by this visualization, is that the final temporal prediction arises from a competition between two distinct and conflicting signals that develop at different stages of processing: an early-stage, content-agnostic structural bias and a late-stage, content-aware semantic grounding.

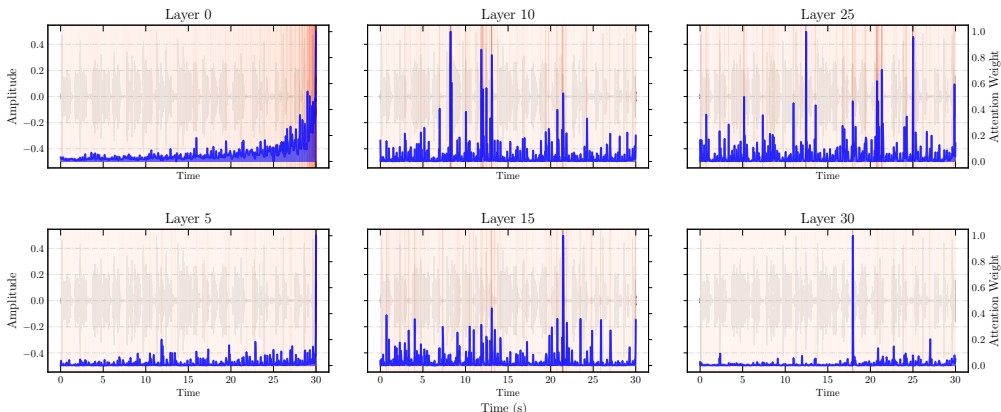

Figure 3: **Evolution of attention distribution across decoder layers.** Each subplot displays the audio waveform (black), attention weights as a heatmap overlay (red), and the attention curve (blue). The six panels show attention for early (0, 5), mid (10, 15), and late (25, 30) decoder layers. (**Early Layers**): Attention exhibits a strong structural bias, disproportionately focusing on the sequence start, a phenomenon we term 'attention front-loading'. (**Late Layers**): The focus gradually shifts from this structural prior to the actual semantic content, with attention peaks correctly aligning with the key speech events.

In the **early layers** of the decoder, such as Layers 0 and 5, the model exhibits a powerful **structural bias**. As depicted in the left panels of Figure 3, the attention pattern is not driven by the audio's semantic content. Instead, it is heavily concentrated at the very end of the sequence (t ≈ 0s), with a smaller echo of attention at the end. This content-agnostic pattern functions as a **Structural Prior**, providing a default "anchor" for the model's attention before semantic processing has fully developed. This low-level *Attention Boundary Bias* offers a compelling mechanistic explanation for the U-shaped positional error curve observed in our main experiments, where events near the sequence boundaries were most prone to localization errors.

This initial pattern undergoes a significant transformation as we proceed to the **late layers**. The middle and right panels of Figure 3 (Layers 25 and 30) show a starkly different attention landscape. The strong boundary bias has largely dissipated, replaced by sharp, localized attention peaks that appear in the middle of the audio stream. This emergent pattern represents **Semantic Grounding**. It is a high-level, content-driven signal that enables the model to correctly understand *what* is being said. For instance, the prominent attention peaks in the later layers align precisely with the ground-truth timestamps of key phrases identified by Whisper, such as "thesaurus handy" and "finished editing".

The synthesis of these observations reveals that attention within LALMs is not a monolithic process but a dynamic evolution. The model simultaneously develops two competing signals: a strong, low-level **structural signal** from the early layers that pulls focus towards the sequence boundaries, and a precise, high-level **semantic signal** from the late layers that correctly identifies the content's temporal location. We posit that the temporal bias observed throughout our experiments is the direct result of the interference and competition between these two signals. When the model is tasked with outputting a single, absolute timestamp, the deeply ingrained structural bias from the early layers can contaminate or "pull" the more accurate semantic signal from the later layers. This hypothesis elegantly resolves the central paradox of our findings: the model can accurately report *what* happened because its late-layer semantic signal is correct, but it systematically misreports *when* it happened because its final temporal decision is corrupted by the powerful, early-layer structural bias. This insight suggests that future work on improving temporal fidelity in LALMs should focus on developing mechanisms to mitigate or decouple this detrimental structural prior.

## 5 RELATED WORK

### 5.1 AUDIO EVENT DETECTION AND TIMESTAMPING

Research on temporal understanding of audio has traditionally centered on sound event detection and localization. Early studies on polyphonic detection relied on convolutional and recurrent neural networks that predicted onset and offset boundaries directly from frame-level features (Cakır et al., 2017; Kong et al., 2020). Later work extended this to spatialized soundscapes, introducing tasks that combine detection with localization in realistic multichannel audio environments (Adavanne et al., 2018; Politis et al., 2022a). Recent advances have applied transformers and conformers to improve timestamp accuracy and robustness in noisy and overlapping conditions (Miyazaki et al., 2020; Cornell et al., 2024). These methods explicitly optimize for temporal boundaries, a contrast to large audio language models that are usually trained without such supervision.

### 5.2 AUDIO-LANGUAGE MODELS (LALMS)

In parallel, the rise of audio–language models has expanded the scope of machine listening beyond transcription. Whisper demonstrated that large-scale weak supervision can yield robust speech recognition with approximate segment-level timestamps (Radford et al., 2023). Subsequent instruction-tuned systems such as AudioGPT (Huang et al., 2024), SALMONN (Tang et al., 2023), and Qwen-Audio (Chu et al., 2023) enable open-ended reasoning across modalities, while generative frameworks such as AudioLM (Borsos et al., 2023) model acoustic continuations. These models emphasize semantic understanding and natural interaction, but precise alignment of predicted events with their actual temporal positions remains underexplored.

### 5.3 TEMPORAL PERCEPTION IN MULTIMODAL MODELS

Temporal reasoning has been more extensively studied in video–language models. Prior work on temporal activity localization and video question answering developed benchmarks where systems must ground language queries to specific moments in video (Anne Hendricks et al., 2017; Gao et al., 2017; Lei et al., 2018; Xiao et al., 2021). Follow-up research on audio–visual event localization further highlighted the challenges of aligning multiple modalities in time (Tian et al., 2018). Although these studies illustrate techniques for handling temporal grounding in video, similar benchmarks and analyses are largely missing for the audio-only domain.

### 5.4 BIAS IN GENERATIVE MODELS

Finally, there is a growing body of research on bias and calibration in generative models. Studies on retrieval-augmented generation have revealed position biases that influence how models use evidence depending on its order (Yao et al., 2025). Other work has documented hallucinations in language models (Ji et al., 2023), as well as degradation in long-context reasoning where relevant information is lost in the middle of prompts (Liu et al., 2023). These findings suggest that generative models may exhibit systematic tendencies when handling temporal information. Our study extends this line of inquiry by focusing on temporal bias, a phenomenon that has not been explicitly examined in audio–language models.

## 6 CONCLUSION

In this work, we conducted the first systematic study of temporal bias in Large Audio-Language Models (LALMs). Our experiments reveal that LALMs exhibit systematic yet heterogeneous misalignments in predicting event timings: bias magnitude and direction vary with sequence length, event duration, and event position. We introduced the **Temporal Bias Index (TBI)** to quantify these directional tendencies and linked observed biases to attention mechanisms through interpretability analyses. These findings highlight fundamental limitations in LALMs' temporal reasoning, emphasizing that accurate semantic understanding does not guarantee precise temporal perception. Our study provides both a rigorous evaluation framework and actionable insights for designing temporally robust audio-language models, and it opens avenues for future research into mitigating temporal bias and enhancing the reliability of LALMs in real-world time-sensitive applications.

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

# A    APPENDIX

## A.1    ANALYSIS OF EARLY AND LATE TEMPORAL BIAS IN PREDICTIONS

Table 5 presents a detailed breakdown of early versus late predictions across different models and window lengths. Notably, early predictions (bias $\leq 0$) consistently exhibit more pronounced temporal deviations than late predictions (bias $\geq 0$). For example, at a 120-second window, the Qwen2-Audio-7B-Instruct model shows an early bias of -24.71 seconds, compared to a much smaller late bias of 4.77 seconds. This clear disparity is most evident at the 120-second window, but across other window lengths, early predictions generally exhibit larger biases than late predictions. This trend is observed across all LALMs, where early predictions tend to have significantly larger biases than late ones. While late predictions do occur, they are generally fewer and less severe. This consistent pattern suggests that anticipation is the dominant error mode, with the LALMs struggling more to avoid predicting events too early rather than delaying them.

# B    USE OF LARGE LANGUAGE MODELS

During the preparation of this manuscript, we utilized a large language model (LLM) to assist in improving the clarity, conciseness, and overall readability of the text. The LLM was employed as a writing aid for grammar correction, sentence restructuring, and polishing of the language to ensure the arguments and findings were communicated as effectively as possible. The core ideas, experimental design, results, and analyses presented in this paper are entirely the contributions of the authors.

Table 5: Experiment 1: Effect of audio window length on temporal bias. Values are presented as Sony/TAU dataset results. "Valid" denotes the number of effective samples used for evaluation. "Bias $\leq 0$" represents mean temporal deviation for early predictions (negative values), "Bias $\geq 0$" represents mean temporal deviation for late predictions (positive values), and "Abs. Bias Mean" shows the overall absolute temporal deviation regardless of direction.

| Model | Window (s) | Valid | Bias $\leq 0$ (mean) | Bias $\geq 0$ (mean) | Abs. Bias Mean |
|---|---|---|---|---|---|
| Voxtral-Mini-3B-2507 | 5 | 726 / 363 | −1.10 / −1.02 | 0.68 / 0.35 | 1.01 / 0.97 |
| | 10 | 492 / 308 | −2.06 / −1.88 | 0.91 / 0.55 | 1.81 / 1.79 |
| | 20 | 280 / 265 | −4.04 / −3.94 | 5.03 / 2.83 | 4.99 / 4.12 |
| | 30 | 211 / 243 | −6.49 / −6.25 | 7.75 / 4.38 | 7.54 / 5.76 |
| | 40 | 144 / 174 | −7.21 / −7.31 | 13.23 / 7.90 | 11.24 / 8.36 |
| | 50 | 138 / 150 | −10.89 / −9.13 | 16.66 / 9.48 | 15.07 / 10.47 |
| | 60 | 114 / 143 | −11.90 / −9.67 | 13.26 / 9.78 | 12.80 / 11.15 |
| | 90 | 68 / 100 | −14.70 / −20.18 | 18.00 / 16.23 | 16.96 / 20.57 |
| | 120 | 45 / 70 | −24.09 / −27.89 | 21.34 / 20.76 | 22.97 / 27.05 |
| Qwen2-Audio-7B-Instruct | 5 | 1650 / 1507 | −0.60 / −0.61 | 2.12 / 1.37 | 2.00 / 1.35 |
| | 10 | 893 / 845 | −1.35 / −1.38 | 3.24 / 2.50 | 3.09 / 2.40 |
| | 20 | 470 / 459 | −3.12 / −2.94 | 4.44 / 3.51 | 4.42 / 3.78 |
| | 30 | 320 / 323 | −5.44 / −5.46 | 5.75 / 4.79 | 6.01 / 5.61 |
| | 40 | 232 / 252 | −8.43 / −8.91 | 6.60 / 5.19 | 7.56 / 7.84 |
| | 50 | 196 / 209 | −11.10 / −12.28 | 7.03 / 4.20 | 9.13 / 9.13 |
| | 60 | 154 / 160 | −11.39 / −13.82 | 8.38 / 6.48 | 10.15 / 11.08 |
| | 90 | 90 / 107 | −20.32 / −22.88 | 11.17 / 5.79 | 15.81 / 18.15 |
| | 120 | 54 / 70 | −24.71 / −32.06 | 4.77 / 3.38 | 14.83 / 25.60 |
| Aero-1-Audio | 5 | 1184 / 1412 | −0.76 / −0.85 | 2.61 / 2.83 | 2.48 / 2.70 |
| | 10 | 705 / 719 | −1.85 / −1.78 | 4.10 / 4.36 | 3.85 / 4.08 |
| | 20 | 399 / 377 | −7.16 / −7.09 | 1.68 / 1.53 | 3.94 / 3.98 |
| | 30 | 228 / 156 | −8.98 / −11.66 | 1.99 / 2.30 | 5.05 / 7.72 |
| | 40 | 218 / 139 | −10.10 / −13.60 | 6.20 / 5.29 | 8.02 / 9.77 |
| | 50 | 182 / 145 | −11.63 / −16.14 | 4.25 / 4.33 | 7.56 / 11.01 |
| | 60 | 145 / 126 | −17.10 / −19.38 | 1.89 / 2.02 | 9.97 / 13.47 |
| | 90 | 90 / 93 | −26.57 / −29.50 | 6.40 / 6.70 | 16.93 / 23.44 |
| | 120 | 51 / 44 | −21.16 / −35.12 | 1.09 / 14.75 | 16.06 / 30.03 |
| Kimi-Audio-7B-Instruct | 5 | 1055 / 1202 | −0.52 / −0.51 | 0.22 / 0.18 | 0.61 / 0.59 |
| | 10 | 700 / 643 | −1.36 / −1.18 | 0.44 / 0.38 | 1.44 / 1.30 |
| | 20 | 240 / 261 | −2.54 / −2.51 | 1.74 / 2.21 | 2.89 / 3.26 |
| | 30 | 129 / 67 | −4.21 / −2.57 | 2.34 / 4.57 | 3.92 / 4.83 |
| | 40 | 50 / 74 | −5.16 / −8.34 | 5.50 / 6.70 | 6.07 / 9.25 |
| | 50 | 26 / 16 | −8.94 / −4.81 | 3.33 / 11.55 | 6.35 / 10.53 |
| | 60 | 6 / 12 | −6.75 / −4.84 | 7.70 / 16.28 | 7.07 / 12.18 |
| | 90 | 15 / 13 | −35.62 / −7.03 | 3.73 / 33.90 | 23.11 / 20.52 |
| | 120 | 5 / 3 | −14.00 / - | 5.50 / 45.23 | 10.60 / 45.23 |
| PretrainedSED | 5 | 1173 / 1328 | −0.07 / −0.15 | 0.14 / 0.14 | 0.17 / 0.23 |
| | 10 | 669 / 747 | −0.31 / −0.35 | 0.37 / 0.23 | 0.52 / 0.42 |
| | 20 | 369 / 425 | −0.47 / −0.55 | 0.73 / 0.44 | 0.88 / 0.68 |
| | 30 | 251 / 301 | −0.34 / −0.64 | 1.27 / 0.91 | 1.23 / 1.10 |
| | 40 | 190 / 243 | −0.46 / −0.77 | 1.32 / 0.70 | 1.30 / 0.95 |
| | 50 | 153 / 198 | −0.88 / −1.69 | 1.21 / 1.63 | 1.44 / 2.10 |
| | 60 | 124 / 160 | −0.46 / −1.00 | 2.22 / 1.26 | 2.01 / 1.49 |
| | 90 | 72 / 105 | −1.61 / −2.15 | 3.63 / 2.46 | 3.56 / 2.79 |
| | 120 | 47 / 78 | −0.54 / −4.37 | 1.17 / 3.06 | 1.21 / 3.87 |

