# OpenReview forum: "Lost in Time: Systematic Temporal Bias in Large Audio Language Models"
_ICLR.cc/2026/Conference — Submitted to ICLR 2026_

### Official Review · Reviewer_PaoL · 2025-10-28

**Soundness:** 3
**Presentation:** 3
**Contribution:** 3
**Rating:** 4
**Confidence:** 4

**Summary:**

This paper proposes a systematic and comprehensive analysis of temporal bias in Large Audio Language Models (LALMs). Through experiments, the paper reveals that LALMs consistently predict the temporal occurrence of acoustic events earlier. Through detailed analysis, the paper shows that in LALMs: 1) temporal bias increases with audio length, 2) prediction precision varies across events, and 3) prediction error is lower when an event is in the middle of the audio and higher at both ends. The paper also shows that the temporal bias results from the interaction between attending to the end in earlier layers and attending to the correct position in later layers.

**Strengths:**

This paper is clearly presented. The experiments are comprehensive, well-motivated, and the conclusions are justified by the experimental results.

**Weaknesses:**

My biggest concern regarding the soundness of this paper is whether the LALMs evaluated in this paper have the ability to perform event localization. To my knowledge, Qwen2-Audio, Voxtral-Mini, Kimi-Audio-7B-Instruct, and Aero-1-Audio do not clearly state in their papers that they trained on the task evaluated here (locating audio events and outputting timestamps). I include some evidence from the discussion on Hugging Face. Since all experiments rely on the implicit assumption that these models can perform this task, a crucial experiment is to test whether the models can perform it with sufficient precision. For example, in addition to the current experiments, the authors can evaluate the precision of identifying an unrelated random event. Although this paper shows that later-layer attention can attend to the correct event position, this does not establish that the model can output the position in timestamps. The fact that localization accuracy varies with event type can also be a result of the model learning a prior over event duration or of bias in the dataset.

https://huggingface.co/moonshotai/Kimi-Audio-7B-Instruct/discussions/3
https://huggingface.co/mistralai/Voxtral-Mini-3B-2507/discussions/35

Minor typo and writing issues

1. The title of the paper in the PDF is not the same as in OpenReview. Please correct it.
2. Line 52: It would be better to change “LALMs do not simply anticipate events earlier than their ground truth, as might be intuitively assumed.” to “LALMs do not simply anticipate events earlier [or later] than their ground truth, as might be intuitively assumed.” Even though the result shows the prediction is always earlier, this phrasing is confusing for readers about why it is earlier.
3. Line 150: reference error.
4. Line 269: it would be better to define the delta mathematically to avoid confusion.
5. It is confusing to me for the results related to Figure 3. It seems the earlier layers prefer attention at the end of audio. Why in line 407 the paper states that “Instead, it is heavily concentrated at the very end of the sequence ($t \approx 0\text{ s}$)”?

**Questions:**

See above.

---

### Official Review · Reviewer_2sLs · 2025-11-01

**Soundness:** 2
**Presentation:** 2
**Contribution:** 1
**Rating:** 2
**Confidence:** 4

**Summary:**

The paper systematically explores the capability of Large audio language models to predict the temporal location of sound events. The analysis is done across datasets, varying audio lengths, and across event types and position in audio.

**Strengths:**

The paper works on an important and interesting task of accurately predicting the temporal location of sound events.  The paper is well written and explores the impact of datasets, audio, and event length on the temporal prediction performance.

**Weaknesses:**

Temporal Bias Index (TBI) is a poorly defined metric. What if some of the errors are positive and some are negative, and the overall TBI is close to zero? This does not mean the model has no bias, as TBI would imply.
Why is the TBI, one of the main contributions of the paper, not used as the primary metric in the paper? The majority of the results use MAE.

**Questions:**

Line 418: “For instance, the prominent attention peaks in the later layers align precisely with the groundtruth timestamps of key phrases identified by Whisper, such as ”thesaurus handy” and ”finished editing”.” How accurate is the alignment between attention peaks and ground truth timestamps?

From section 4 interpretability analysis, lines 426-427
“When the model is tasked with outputting a single, absolute timestamp, the deeply ingrained structural bias from the early layers can contaminate or ”pull” the more accurate semantic signal from the later layers.” if the early layers contaminate the signal from the later layers then why does it not affect other tasks such as ASR or speech to speech translation which also require some temporal information?

Currently, there is not enough evidence to support the claim. It is unclear if the structural bias in the early layers is the cause of poor absolute timestamps prediction, or if some correlation exists between the two.

---

### Official Review · Reviewer_fbAp · 2025-11-03

**Soundness:** 2
**Presentation:** 2
**Contribution:** 1
**Rating:** 2
**Confidence:** 4

**Summary:**

The paper presents the first systematic study of temporal bias in Large Audio Language Models (LALMs), exposing a fundamental limitation in their ability to accurately determine when events occur within an audio stream. Temporal awareness—knowing *when* an event happens—is essential in real-world tasks such as lecture indexing or analyzing political debates. The authors identify temporal bias as a consistent tendency for LALMs to misplace audio events along the time axis. While these models demonstrate strong semantic understanding by summarizing key events accurately, they often misreport timestamps by several seconds. For example, when locating a formula introduction in a lecture, models tend to predict systematically earlier or later timings than the ground truth—sometimes by more than seven seconds—even though their attention mechanisms correctly focus on the relevant segments.

To explore this issue, the researchers conducted controlled experiments manipulating three key factors: sequence length, event duration, and event position, using the STARSS22 dataset. They introduced a new metric, the Temporal Bias Index (TBI), defined as the average signed difference between predicted and actual onset times.

A negative TBI indicates systematic early bias, while a positive value indicates late bias. The Mean Absolute Error (MAE) complements this by measuring the magnitude of deviation regardless of direction. Four state-of-the-art LALMs—Voxtral-Mini-3B-2507, Qwen2-Audio-7B-Instruct, Kimi-Audio-7B-Instruct, and Aero-1-Audio—were evaluated against a non-LALM baseline, PretrainedSED (Sound Event Detection).

The findings revealed that temporal bias is systematic yet varies across models, pointing to deeper architectural issues. First, the bias is pervasive: all LALMs tested displayed consistent patterns across different domains (acoustic events, conversations, music), with TBI values between -4.7 and -6.8 seconds, whereas the supervised baseline maintained near-zero bias. Second, audio length significantly amplifies the bias—errors grow non-linearly with longer inputs. In 120-second contexts, MAE values rose to over 20 seconds for some models, reflecting both increased bias and variance. Notably, models exhibited a stronger tendency toward early predictions as context length increased. Third, event duration and nature influenced performance: longer events worsened localization accuracy, and transient sounds (e.g., knocks, bells) caused greater degradation than sustained speech. The deviation increase for transient events, such as “Door Closing,” was especially pronounced. Fourth, event position introduced distinct patterns—Voxtral displayed a symmetric, U-shaped error curve with higher errors at clip boundaries, while Qwen2-Audio showed asymmetric bias with larger errors toward the end of clips.

To explain why this bias occurs, the paper connects it to the models’ internal attention mechanisms. Temporal predictions arise from the interplay between two signals developing across decoder layers. Early layers exhibit a structural prior or “attention front-loading,” where attention is disproportionately concentrated near the beginning of the sequence (around 0 seconds). This default bias acts as a temporal anchor, accounting for boundary-related errors. Later layers, however, produce content-aware semantic grounding, focusing accurately on the true event. The observed temporal bias emerges when these two signals interfere—accurate semantic cues are distorted by the ingrained structural bias from earlier layers. In other words, while the models understand *what* happens, their internal timing mechanisms misrepresent *when* it happens.

The authors liken this to a baker who knows every ingredient in a cake but whose timer is unreliable—starting too early or finishing too late depending on the oven’s size, not the recipe itself. Similarly, while LALMs excel at semantic comprehension, their temporal foundations remain fragile, revealing a critical gap in the alignment between audio understanding and temporal precision.

**Strengths:**

The paper stands out for its novelty, methodological rigor, and deep interpretability analysis of Large Audio Language Models (LALMs). Its primary strengths lie in being the first to systematically explore temporal bias in LALMs, introducing a new framework for its quantification, and uncovering valuable insights into the internal mechanisms driving this bias.

First and foremost, the study pioneers the systematic investigation of temporal bias in LALMs, addressing a major research gap. While prior work has focused on temporal reasoning in multimodal or video-language models, no equivalent effort has been made for audio-only models. By isolating and quantifying how these models misalign temporally, the paper introduces a new research direction for understanding generative biases in audio-based AI systems.

Another major contribution is the creation of a robust evaluation framework and a novel metric, the Temporal Bias Index (TBI). This metric measures the average signed deviation between predicted and actual event onset times, thereby capturing whether models consistently anticipate or lag behind true timestamps. Through controlled experiments, the authors systematically examine how sequence length, event duration, and event position contribute to bias, allowing a detailed and reproducible analysis of the phenomenon.

Empirically, the paper delivers comprehensive and cross-model findings across multiple leading LALMs—such as Voxtral-Mini-3B-2507, Qwen2-Audio-7B-Instruct, Kimi-Audio-7B-Instruct, and Aero-1-Audio—using diverse datasets that span speech, acoustic events, and music. The results show that temporal bias is a pervasive and consistent problem, with models accurately capturing what happens but consistently failing on when it happens. Moreover, bias severity scales non-linearly with input length and varies across event types and positions, particularly at audio boundaries, revealing a heterogeneous and structurally embedded issue.

A particularly strong aspect of the work is its mechanistic interpretability analysis, which uncovers the internal cause of these biases. Using layer-wise attention visualizations, the authors show that early decoder layers exhibit a structural bias that “front-loads” attention toward the beginning of the sequence, while later layers provide accurate semantic grounding. The resulting timestamps reflect a contaminated synthesis of these two conflicting signals—explaining why models can understand content correctly yet misreport time.

Finally, the paper provides clear implications for future model design. It identifies temporal bias as a core architectural limitation and calls for the development of temporally robust architectures that disentangle structural priors from semantic understanding. This insight has direct relevance for time-sensitive applications such as lecture indexing, audio-visual synchronization, and event retrieval.

In summary, the paper serves as both a diagnostic framework and a wake-up call for the LALM community: while these models excel at interpreting and generating content, they remain fundamentally unreliable at aligning that content in time—functioning as expert narrators but flawed chronometers.

**Weaknesses:**

The paper investigates a fundamental flaw in Large Audio Language Models (LALMs): their consistent inability to maintain temporal precision despite strong semantic understanding. The study reveals that while these models can correctly interpret and summarize events in audio data, they frequently misreport the exact timing of these events—sometimes by several seconds—highlighting a critical dissociation between semantic comprehension and temporal localization. This issue becomes more severe as audio length increases, with mean absolute errors expanding non-linearly in longer contexts. For instance, the temporal error grows to over 20 seconds for extended 120-second audio inputs, reflecting both increasing bias and variability. Furthermore, the paper identifies anticipation as the dominant error mode, where models tend to predict events earlier than they occur. Temporal bias also varies by event type and position: transient sounds like knocks or bells exhibit greater deviations than speech, and events near audio boundaries are especially error-prone. The root cause is traced to an architectural flaw within LALMs—the interference between low-level structural biases that overemphasize sequence boundaries and higher-level semantic signals that correctly identify content. This contamination leads to systemic timestamp inaccuracies. Ultimately, the paper concludes that LALMs suffer from “temporal naivety,” where their architecture’s design, particularly in positional encoding and attention mechanisms, hinders precise time alignment. This limitation renders current models unreliable for real-world, time-sensitive audio applications that demand both semantic and temporal accuracy.

**Questions:**

Thanks for the work, looking forward towards making the publication better. kindly answer the following questions.

1. [Figure 1] : Which plot are you showing on the right, with the attention weight vs time? Which attention weight does this layer belong to? Also how are you constructing the map between Audio-Sound-Wave and Attention weight?

I am a bit confused as you have audio summarization task on the left, how does the audio-summarization task produce attention weights with respect to time?

2. [Line 81-88] : In each of these points, kindly add the reference to section where you've addressed these contributions? Also importantly show what do you mean by the terms on [Line 86], like bias to attention, bias to positional encoding compression, temporal robustness. Keeping them vague at the start, makes it difficult for the readers to follow.

3. [Line 93] : "initial experiments" which experiment? What did it identify? Should take care while writing the article. Disturbs the flow, you can easily put a reference to the section.

4. [Line 97] : "similar pattern" -> which pattern? (you just wrote an example before this line, not a general trend/ pattern); "this systematic bis" -> which bias?

5. [Line 122] : Ummm, so you don't wish to consider the sign of the temporal difference? So for TBI what if in a particular audio there are two events, event A and B. Event A leads by 1 sec and Event B lags by 1 sec; the TBI = 0 ; right?

6. [Line 155-159] : These are actually good questions, can you please formalize these as of now this section looks a bug vague. Follow this up in your experimental section. (May include a figure which helps the reader [optional]). The whole section 3 needs a lot of re-writting, the concepts look good but there is no way a paper written like this can be promoted towards good scores at ICLR. Suggestion, don't be descriptive, be focussed and stay away from LLM based write-ups. As of now it looks like someone copied the results table and asked an LLM to write these sections. (The reviewer is not flagging this as lack of ethics, but hopes that the authors will address these issues).

7. [Section 5] : Related works : I recommend you to cite few other papers which have also looked in the field of temporality of Audio-language models (Have previously reviewed these work) : 1) Sinha, Anshuman, et al. "Enhancing Audio-Language Models through Self-Supervised Post-Training with Text-Audio Pairs." arXiv preprint arXiv:2408.09269 (2024)., 2) Ghosh, Sreyan, et al. "Compa: Addressing the gap in compositional reasoning in audio-language models." arXiv preprint arXiv:2310.08753 (2023).

The paper needs a lot of formalism, also the conclusion looks a bit weak for such an avenue which addresses representation learning (as the name suggests ICLR). Kindly address all the above points, looking forward towards the discussion. All the best.

**Details Of Ethics Concerns:**

No ethics review required,

---

### Official Review · Reviewer_BpbL · 2025-11-10

**Soundness:** 3
**Presentation:** 3
**Contribution:** 2
**Rating:** 4
**Confidence:** 5

**Summary:**

This paper studies temporal localization ability (i.e., temporal bias) of large audio–language models (LALMs) when answering timestamp-based audio queries. The authors conduct experiments over multiple scenarios (audio length, event duration, and event position) to primarily use STARSS22 to evaluate the SED performance via Temporal Bias Index (TBI) and Mean Absolute Error (MAE) metrics on recent LALMs (e.g., Voxtral, Qwen2-Audio, Kimi-Audio, Aero). And the results show that these LALMs are strong in audio-conditioned semantic reasoning, but their ability on sound event detection has been largely unexplored.

**Strengths:**

The strengths of this paper lie in its novelty and experimental design. This paper explores an important limitation of the current LALMs: the sound event detection ability to tell "when" of the sound events apart from "what". The experimental design for this exploration is clear and systematic. The paper first design a simple and intuitive metric (Temporal Bias Index) for directional error analysis. And the analysis decomposes factors affecting timestamp prediction, including audio length, event duration, event position, and provides quantitative evidence for each.

The paper also comprehensively compares multiple models, such as Voxtral, Qwen2-Audio, Kimi-Audio, and Aero, against a supervised SED baseline. This demonstrates the generality of temporal failures of these LALMs. The visualization of hierarchical attention transition is insightful and consistent with the measured biases.

**Weaknesses:**

However, there is one critical and fundamental confound, as weakness, in this paper.

 **All LALMs tested were not trained for SED or timestamp prediction tasks and data**

The paper evaluates LALMs on SED, but none of them were trained for timestamp supervision, and none were instructed or optimized for sound event onset/offset precision. This is not due to the authors' fault, it is just because almost all LALMs are not trained on SED. However, this confound points out some wrong assumptions in this paper: poor MAE/TBI results are not evidence of an intrinsic temporal bias, but simply that the models have never been trained on this task. This severely limits causal interpretation. Under this circumstance, in section 4, the interpretation that temporal failure arises from early-layer structural priors overpowering later-layer semantic grounding is not solid. And the conclusion “LALMs have temporal bias” therefore borders on tautology: models not trained to localize events cannot localize events well.

The claim should be reframed: “LALMs do not naturally acquire SED ability from current training pipelines.” And without fine-tuning, this work cannot distinguish model architecture limitations and prompting limitations of these LALMs.  Many existing findings may reflect task mismatch, not “bias.” And this is the major conceptual weakness of the paper.

However, there are someways to make this paper valuable again, as currently this paper points out the limitation but focus on wrong direction. There *are* some audio foundation models (perhaps not LALMs) that focusing on the temporal bias. It is better if the paper could:
1. Compare such models (FLAM [1], T-CLAP [2], and CLAP [3] as an low anchor) to a pretrained SED model.
2. To establish whether the deficit of temporal bias in LALM is architectural or merely training-data dependent, the paper must include some LALMs with lightweight fine-tuning (e.g., LoRA) for SED tasks.

Both of them would help determine whether LALMs can learn temporal alignment, or LALMs lack internal temporal structure regardless of supervision even with finetuning.

Without these comparisons, the paper is incomplete.

And there is a minor but also critical issue: the paper title in the openreview is different from the title in the submission. This is usually forbidden and could cause desk-reject.

[1] FLAM: Frame-Wise Language-Audio Modeling

[2] T-CLAP: Temporal-Enhanced Contrastive Language-Audio Pretraining

[3] Large-scale Contrastive Language-Audio Pretraining with Feature Fusion and Keyword-to-Caption Augmentation

**Questions:**

1. Have the authors finetuned any LALMs for SED or timestamp prediction? Even 1–2 LoRA finetuned models would clarify whether this issue stems from training or from architecture.

2. How do some audio foundation models focusing on temporal bias perform? Models such as FLAM, T-CLAP, or existing CLAP.

3. Can prompting change outcomes? Did structured prompting (format constraints, step-by-step reasoning, forced timestamp search) help on LALMs before and after funetuning?

---

### Meta-Review · Area_Chair_DnBt · 2026-01-05

**Summary:**

There are two main concerns from the reviewers. The first (raised by reviewer BpbL and PaoL) is that the models tested are not meant to localize sound events. Any findings for these models are uninteresting by nature. The other problem is the TBI metric, as pointed out by fbAp and 2sLs. A small TBI value could be the result of averaging positive and negative values, not because that the errors are small. The problems are serious enough that the experiments (including the design) will need to be revamped.

**Reviewer Concerns:**

No rebuttal submitted.

**Reviewer Scores:**

Not applicable as there is no rebuttal.

---

### Decision · Program_Chairs · 2026-01-26

Reject